# Development of Intracorporeal Differentiation of Stem Cells to Induce One-Step Mastoid Bone Reconstruction during Otitis Media Surgeries

**DOI:** 10.3390/polym14050877

**Published:** 2022-02-23

**Authors:** Sung-Hee Park, Hantai Kim, Yun Yeong Lee, Yeon Ju Kim, Jeong Hun Jang, Oak-Sung Choo, Yun-Hoon Choung

**Affiliations:** 1Department of Otolaryngology, Ajou University School of Medicine, Suwon 16499, Korea; soyclara99@naver.com (S.-H.P.); noto.hantai@gmail.com (H.K.); seven260@ajou.ac.kr (Y.Y.L.); yeonju0130@naver.com (Y.J.K.); jhj@ajou.ac.kr (J.H.J.); 2Department of Otorhinolaryngology, Eulji University School of Medicine, Uijeongbu 11759, Korea

**Keywords:** mastoidectomy, osteogenesis, stromal vascular fraction cells, polycaprolactone (PCL) scaffold, autologous growth factor

## Abstract

Mastoidectomy is a surgical procedure for the treatment of chronic otitis media. This study investigated the ability of rat stromal vascular fraction cells (rSVF) in combination with polycaprolactone (PCL) scaffolds and osteogenic differentiation-enhancing blood products to promote the regeneration of mastoid bone defect. Twenty male Sprague Dawley rats were randomly divided according to obliteration materials: (1) control, (2) PCL scaffold only, (3) rSVFs + PCL, (4) rSVFs + PCL + platelet-rich plasma, and (5) rSVFs + PCL + whole plasma (WP). At 7 months after transplantation, the rSVFs + PCL + WP group showed remarkable new bone formation in the mastoid. These results indicate that SVFs, PCL scaffolds, and blood products accelerate bone regeneration for mastoid reconstruction. Autologous SVF cells with PCL scaffolds and autologous blood products are promising composites for mastoid reconstruction which can be easily harvested after mastoidectomy. With this approach, the reconstruction of mastoid bone defects can be performed right after mastoidectomy as a one-step procedure which can offer efficiency in the clinical field.

## 1. Introduction

The management of middle ear disease occasionally requires a canal wall-down mastoidectomy. Mastoidectomy is a surgical procedure to remove an infected mastoid bone and air cells that have a honeycomb-like framework. The inner surfaces are also covered with mucosa resulting from chronic ear diseases such as otitis media and mastoiditis. This structure is associated with gas exchange function and pressure regulation of the middle ear [1,2,3]. Although this surgical procedure is indispensable for the eradication of chronic ear diseases, it has a common problem of altering the anatomy and physiology of the middle ear which can cause repetitive chronic infections, dizziness, granulation tissue formation, and the accumulation of keratin debris [4]. Furthermore, open cavity mastoidectomy may also cause difficulty in fitting a hearing aid as well as vertigo attacks following pressure or temperature changes and loss of the ability to self-clear the external acoustic canal [5]. To reduce open cavity problems, mastoid obliteration along with the reconstruction of the posterior wall of the external auditory canal and air cells has been attempted [6,7,8,9].

Autologous grafts are preferred over alloplastic and biosynthetic materials because of biological safety, cost effectiveness, and the provision of an ideal construct supplying osteoinductive growth factor. However, these autologous tissues also have limitations such as insufficient amount, restricted donor site defects, and unpredictable resorption after surgery [10,11]. To overcome these problems, we used uncultured stromal vascular fractions (SVFs), which can be transplanted without further in vitro selection or expansion steps; possess the multiple capacity to differentiate into adipose osteoblasts, adipocytes, chondrocytes, and myocytes rapidly, as compared to bone marrow stem cells; and have the advantage of providing a non-invasive harvest approach [12]. Recent evidence showed that an SVF also induced osteogenic differentiation with a hydroxyapatite scaffold [13] and there was a minor difference in kinetics and phenotype of differentiation between freshly isolated adipose stem cells (ASCs) and cultured ASCs [14]. Stromal cells from adipose tissue have a higher proliferative rate, autocrine production of growth factors, and immunomodulatory effects to reduce the immune response for tissue regeneration [15]. Moreover, SVF cells cultured within 3D ceramic scaffolds induced osteogenesis after subcutaneous implantation in vivo [16]. The clinical application of free fat isolated from liposuction aspirates shows an alternative strategy to soft tissue augmentation surgery to tissue reconstruction [17]. In fact, SVF enrichment was proven to stimulate the secretion of growth factors encouraging angiogenesis, as well as pericytes and endothelial cells, contained in the stromal vascular fraction [18,19].

Polycaprolactone (PCL) nanofiber scaffold is a biodegradable polyester material, which has been widely employed to support adhesion, proliferation, and acceleration of osteogenic differentiation in MSC-based cell regenerative therapy for treating bone impairment [20,21,22,23]. PCL nanofiber scaffold has also been exploited in several kinds of adhesion barriers, implants, and drug delivery devices that have been approved by the FDA. Recent in vivo studies evidently showed that the molecular weight of explanted PCL material in rabbit calvaria had decreased by 63% in 2 years [24] and implanted PCL material in the dorsal skin of rat was metabolized by unknown process which ultimately excreted it from the body and this was detected at day 15 [25].

The utilization of autologous growth factors derived from blood can represent an effective support in tissue regeneration due to their ability to stimulate cell proliferation, differentiation, and neo-angiogenesis, and was investigated for clinical use [26]. The widely studied platelet-rich plasma (PRP), obtained by blood centrifugation, is widely applied in clinical areas such as orthopedics, ophthalmology, and otorhinolaryngology. PRP has various growth factors, including fibroblast growth factor, epidermal growth factor, vascular endothelial growth factor, platelet-derived growth factor, transforming growth factor-β, and insulin-like growth factor [27]. Despite the focused use of PRP, the efficacy of PRP remains under debate because of the lack of consensus in the optimal preparation condition for tissue regeneration [28]. The clear effect of PRP on osteogenic differentiation remains to be elucidated and some studies have reported conflicting results about the effect of PRP on osteogenesis [29,30].

Previous studies have often compared the abilities and functions of PRP and platelet-poor plasma (PPP) [27], but rarely compared these with whole plasma (WP), a bioproduct obtained from pretreatment. Given the unstandardized protocol and the inconsistent results of PRP in clinical applications, WP is expected to be a stable clinical material as it escapes the risk of high leukocyte involvement and the excessive secretion of pro-inflammatory cytokines, and is obtained by a simple one-step manipulation.

In this study, we investigated the biological effects of uncultured autologous adipose tissue-derived SVFs and blood products on a PCL scaffold on osteogenic differentiation. Through this approach, we tried to develop a technique that can be performed in a convenient and efficient one-step procedure immediately after mastoid resection by confirming the anatomical and functional reconstruction of mastoid bone defect.

## 2. Materials and Methods

### 2.1. Fabrication of the PCL Scaffold

The PCL scaffolds were fabricated via a solvent-casting/salt-leaching technique. Briefly, PCL (Sigma-Aldrich, St. Louis, MO, USA) was dissolved in tetrahydrofuran (10% *v*/*v*, Sigma-Aldrich, St. Louis, MO, USA). NaCl (Amresco, Solon, OH, USA) was added to the PCL solution and the suspension was continuously stirred until the viscosity increased. The final mixture was cast into cylindrical acryl molds and dried in a fume hood for 24 h after which the solvent was completely evaporated. Finally, the scaffolds were immersed in distilled water for 2 h to remove NaCl and then dried for 24 h.

### 2.2. rSVF Cells Isolation and Culture

SVFs were isolated from Sprague Dawley (SD) male rats (7 weeks) according to previous reports [31,32]. Briefly, 200 g of fat from subcutaneous adipose tissue was removed under sterile conditions, washed with PBS containing 1% *v*/*v* penicillin/streptomycin, and finely minced with blades. The chopped fat was incubated with low-glucose Dulbecco’s modified Eagle’s medium (DMEM), which contained 0.075% collagenase type I (Sigma-Aldrich, St. Louis, MO, USA), at 37 °C for 30–60 min. The solution was then neutralized with low-glucose Dulbecco’s modified Eagle’s medium (DMEM) containing 10% fetal bovine serum (FBS) and filtered through 100 µm nylon mesh strainers. The high density fraction was then sedimented after centrifugation at 1300× *g* for 3 min. The pellet was resuspended in DMEM, which was then cultured in a humidified incubator under 37 °C in an atmosphere containing 5% CO_2_ in basal media (DMEM, 10% FBS, 1% *v*/*v* penicillin/streptomycin). Culture media were changed every 2 days before use. Upon reaching about 80% confluence, the cells were trypsinized and replated at 1.0 × 10^5^ cells/60 millimeters-squared dish for further study. Cells from the first passage were used individually.

### 2.3. Fluorescent-Activated Cell-Sorting Analysis

Day 1 and 6 of passage 0 rSVF cells were detached with 0.25% trypsin/EDTA (Corning) and centrifuged at 1300 rpm for 3 min. The supernatant was discarded and the rSVF cells were washed using PBS, centrifuged at 1300 rpm for 3 min, and the pellet resuspended in sterile FACS buffer (PBS, 1% FBS) containing fluorescein isothiocyanate (FITC) conjugated CD29 and CD90 (BioLegend, San Diego, CA, USA) for 1 h at 4 °C. The analysis was carried out with a BD FACSAriaTM III (BD Biosciences, San Jose, CA, USA).

### 2.4. Preparation of Rat Blood Products

Blood products were prepared from SD male rats (7 weeks) used for SVF isolation. In brief, whole blood was drawn, via puncture of the heart, into a syringe. The whole blood sample from SD rats was initially collected into a 5.4 mg (spray-dried) K2 EDTA tube (BD Vacutainer^®^, Becton Dickinson, NK, USA) which also functioned as an anticoagulant. The whole blood sample was centrifuged for 25 min at 400× *g* at 4 °C to separate the plasma containing the platelets from the red blood cells. After the formation of three layers, only the WP containing buffy coat was transferred to a new tube and mixed well. The 10% volume of the whole blood was transferred to the new tube and the rest of the whole plasma was further centrifuged for 15 min at 800× *g* at 4 °C to separate the platelets. Platelet-poor plasma, which comprised 10% of the volume of the whole blood, was drawn off the top and PRP, which comprised 10% of the volume of the whole blood, was drawn off the bottom. Then, platelets were activated by 10% (*v*/*v*) CaCl_2_ for 1 h at 37 °C and 12 h at 4 °C to induce the exocytosis of α-granules and growth factors. After clot formation, blood products were centrifuged at 3000× *g* for 20 min and the supernatant was aspirated and then filtered through 0.22 µm filters.

### 2.5. Osteogenic Differentiation

To induce osteogenic differentiation, the first passage rSVF cells were seeded into a 60 mm^2^ dish at a density of 1.0 × 10^5^ cells and incubated at 37 °C in an atmosphere containing 5% CO_2_. Subsequent to adding the cell suspension with osteogenic differentiation media containing low-glucose DMEM (10% *v*/*v* FBS, 1% *v*/*v* penicillin/streptomycin) supplemented with L-ascorbic acid 2-phosphate (50.0 µM final), dexamethasone (0.1 µM final) and β-glycerophosphate (10.0 mM final) to the cell culture dish, the culture media were replaced every 2–3 days.

### 2.6. Alizarin Red S Staining

The passage 1 SVFs were cultured in each blood production medium with low-glucose DMEM for 21 days. Then, the cells in each treatment group were stained with Alizarin Red S solution (Sigma-Aldrich, St. Louis, MO, USA) according to the manufacturer’s instructions to determine the presence of calcium crystal deposition, an indicator of osteogenic differentiation. This is an early stage marker of matrix mineralization, an essential step toward the formation of calcified extracellular matrix associated with bone. At each analysis, SVFs were fixed in 4% paraformaldehyde for 30 min, followed by normal saline washing and staining with 2% Alizarin Red S (pH 4.2) for 10 min at room temperature. Next, 20% methanol 10% acetic acid solution was added to the stained rSVFs, which were then left for 30 min. The mineral content was determined by absorbance values at 450 nm using a microplate reader [33].

### 2.7. In Vivo Animal Study

Twenty SD male rats (7 weeks, 200–250 g) were used in this study. Ajou University School of Medicine—Institutional Animal Care and Use Committee (IACUC number, 2019-0049) approved the surgical procedures in accordance with the guidelines regarding the care and use of animals for experimental procedures. All efforts were made to minimize the number of animals used and their suffering. The animal models for mastoidectomy followed our previous study [34]. Briefly, SD rats were anesthetized with an intraperitoneal injection of Zoletil 50 (0.1 cc/100 g; Virvac Laboratories, Carros, France) and 2% Rompun (0.2 cc/100 g; Bayer Korea, Ansan, Korea). For mastoid bulla defects, the anterior midline neck skin was incised using a scalpel to expose both sides of the bulla (anterior approach), and 3 × 3 mm holes were made using a straight pick. Then, 3D PCL scaffold was placed in the mastoid cavity in the PCL group. Fat (50 mg) was transplanted directly into the 5 mm diameter of the PCL scaffolds with 50 µL of normal saline and placed in the bulla cavity in the PCL + fat (PF) group. The remaining groups were transplanted with 50 µL of blood product components such as PRP and WP, respectively (Figure 1).

We allocated 4 SD rats to each group. However, one in the control group and two in the PFP group died before the evaluations at 7 months. Thus, final radiologic (micro-CT) and histological (immunohistochemistry) studies were performed on a total of 17 rats.

### 2.8. Radiologic Evaluation

The newly formed bone was analyzed after imaging the bulla samples using micro-CT (SkyScan 1076; SkyScan, Kontich, Belgium), as in previous studies. Briefly, a 35 kV source voltage and a 170 μA source current were used, and 569 images per bulla sample were obtained with an image pixel size of 36.44 μm with a 1.0 mm aluminum filter. Images obtained as individual sections were modeled in 3D using 3D image-processing software (MIMICS 16.0; Materialize Mimics, Leuven, Belgium). The newly formed bone was identified using the bone mineral content within each bulla. Then, the bone volume and the bone/total volume were calculated from the micro images for quantitative comparison between groups.

### 2.9. Histological Study

After 7 months, temporal bones were fixed with 4% paraformaldehyde and kept overnight at 4 °C. After decalcification with Calci-Clear Rapid solution (National Diagnostics, Hs-105) for 10 days, temporal bones were embedded in paraffin and 10 micrometers-thick sections cut, deparaffinized, rehydrated, and then subjected to modified antigen-retrieval using pepsin and non-specific peroxidase blocking. Some specimens were stained with hematoxylin (Youngdong Pharmaceutical Co., Seoul, Korea). Bright-field microscopy images were obtained using Picture Frame software (Olympus Optical, BX51, Tokyo, Japan). Osteogenesis was evaluated by immunohistochemical staining. The specimens were reacted with a primary antibody against osteocalcin (Abcam, Cambridge, MA, USA) or bone sialoprotein (Proteintech, Rosemont, IL, USA). To quantify the extent of cellular integration into the scaffold, 2–4 fields of view (200-fold magnification) were chosen at random and analyzed by ImageJ software (Wayne Rasband, National Institute of Health, Bethesda, MD, USA) as reported previously [35].

### 2.10. Statistical Analysis

One-way analysis of variance (ANOVA) with post hoc Bonferroni test was performed between all groups, except for the analysis of the measurement by micro-CT. The Mann–Whitney *U* test was used to identify statistical significances for the measurement of newly formed bone by micro-CT. In all analyses, a *p*-value < 0.05 indicated a statistically significant difference. All statistical analyses were performed using IBM SPSS Statistics for Windows (version 23.0; IBM Corp., Armonk, NY, USA).

## 3. Results

This section is divided by subheadings. It provides a concise and precise description of the experimental results and their interpretation, as well as the conclusions that can be drawn.

### 3.1. Characterization of rSVF Cells

To investigate the surface markers of SVFs obtained from 7-week-old male SD rats, flow cytometry analysis was performed, and the results clearly demonstrated the ability of rSVF cells in passage one to differentiate under the conditions in this study. On day 1 of the culture, the rSVF cells were positive for CD29 (5%) and CD90 (2.5%). The cultured rSVF cells were positive for CD29 (40.8%) and CD90 (40.3%) by day 6 (Figure 2B). Altogether, the results demonstrated that rat SVFs had MSC-specific phenotype characteristics and possessed differentiation potential to form osteogenic lineage cells.

### 3.2. Phenotype of rSVF Cells Treated with Blood Products in Osteogenic Differentiation

During passage zero, rSVFs showed high colony-forming unit frequencies and characteristic spindle-shaped and fibroblast-like morphologies (Figure 2A). The differences between the groups were apparent after three days from treatment. Increased proliferation rate and collagen fibers were observed around the extracellular matrix in the rSVFs treated with blood products compared with the 10% FBS control. Within a week, hydroxyapatite was generated in-between rSVF cell populations. In the PPP and the FBS (control) groups, both the number and the size of mineral nodules were small. In the PRP group, the number of mineral nodules was small, but the size was about 100 µm. On the other hand, the degree of extracellular mineral accumulation in the WP group was significant (Figure 3).

### 3.3. Semi-Quantitative Analysis of Alizarin Red S Staining

Elongated clusters of rSVFs were collected from cell cultures which were treated with blood products (10% of whole volume) followed by Alizarin Red S staining (Appendix A). This anthraquinone dye staining assay was employed to follow the mineralization of differentiated rSVF cells and was performed 3 weeks after cell transduction, showing bright red-colored mineralized matrix in treated cells (Figure 4A). Interestingly, rSVF cells treated with WP produced significant mineralized deposits (1.47 ± 0.05) (Figure 4B), demonstrating that WP has a more prominent possibility of providing osteoinductive and accessible blood derivatives than PPP or PRP that require two-step centrifugations at risk of expose. As shown in Figure 4B, calcium depositions of the control (0.50 ± 0.01), PPP (0.78 ± 0.01), and PRP (0.51 ± 0.01) groups were not as efficient.

### 3.4. Evaluation of New Bone Formation by Micro-CT

After 7 months, based on quantitative assessment by micro-CT (Figure 5A), a significant amount of regenerated bone volume versus total volume (BT/TV) was observed in the PCL (7.72% ± 3.29) and the PCL scaffold + fat + WP (6.32% ± 2.06) groups, consistent with calcium deposit staining. In contrast, the PCL scaffold + fat (2.15% ± 0.88) and PCL scaffold + fat + PRP (2.02% ± 1.14) groups had relatively small amounts of bone formed on micro-CT images (Figure 5B).

### 3.5. Evaluation of New Bone Formation by Immunohistochemistry (IHC)

The effect of rSVF cells loaded on PCL scaffold and blood products on induced osteogenic differentiation in vivo was also evaluated by using immunohistochemical staining of bone sialoprotein and osteocalcin expressed, which are markers of osteoblasts (Figure 6A). Positive osteocalcin expression was significant in the PCL + fat + WP (PFW) group and the PCL group, 111.20 (±0.54) and 86.36 (±7.23), respectively (*p* < 0.005) (Figure 6B). In addition, the rSVF loaded on scaffold with and without PRP groups showed less osteogenesis; 67.73 (±11.68) and 61.00 (±0.765), respectively.

## 4. Discussion

Reconstruction of the mastoid bone defect caused by mastoidectomy is an orchestrated event. Clinically, it is necessary to accelerate bone regeneration after surgery. Conventional methods of healing temporal bone defects, such as bone grafts and grafting materials, cause severe morbidity and bone reconstruction is not always completed [36]. Therefore, this study developed a convenient and efficient one-step technique that can be performed immediately after mastoid resection by confirming bone reconstruction in the mastoid bone defect. We first showed that rSVFs can form calcium mineral nodules in three blood products: whole plasma (WP), platelet-poor plasma (PPP), and platelet-rich plasma (PRP) (Figure 3). Quantification of the degree of mineralization showed remarkably higher osteogenic differentiation with WP than with PPP or PRP (Figure 4). To investigate bone formation, micro-CT was used to assess the bone tissue in the SD rat mastoid temporal bone 7 months after surgery. Similar to the in vitro result, WP-treated rSVFs with PCL scaffold resulted in the most bone reconstruction.

Various bone tissue engineering approaches have been used to reconstruct bone. Facilitating biomaterials provide a favorable environment for bone regeneration [37]. The biological augmentation of scaffolds by cell colonization increases osteoinduction remarkably. The main purpose of using scaffolds is the retention and delivery of cells and biochemical factors for cell attachment and migration. They also serve as templates to guide the development of new tissues. Materials used for scaffolds include natural (protein-based natural biomaterials, silk, collagen, fibrin, hyaluronan, alginate, agarose, and chitosan) and synthetic (ceramics and polymers) biomaterials. Synthetic biomaterials can serve as scaffolds for culturing stem cells, control the mechanical properties and degradation rate, and can be shaped independently [38]. Ceramic scaffolds include those made of calcium sulfate, calcium phosphate, tricalcium phosphate, and hydroxyapatite. These materials do not promote osteoinduction or osteogenesis. Instead, they maintain osteoconduction by providing a structure and framework for the growing bone tissues. Calcium sulfate is absorbed most rapidly at 4 to 12 weeks. Hydroxyapatite is absorbed slowly and can be seen on radiographs for more than 10 years. Calcium phosphate and tricalcium phosphate are also absorbed slowly, this process taking 6 to 24 months [39,40,41,42]. Ceramic bone replacement products are very diverse, but only a few large clinical trials and randomized controlled studies have been conducted [43]. Slowly degrading polymer scaffolds are also being studied for tissue engineering. Polycaprolactone (PCL), a polyester that is resorbed slowly over 2–4 years, is a good candidate for regenerative applications. It is elastic and consists of nonpolar methylene groups and one semi-polar ester group. In tissue engineering, PCL has marked potential for bone regeneration and sufficient mechanical properties to tolerate stress loads after implantation. It is also used widely in various drug delivery applications and has obtained FDA approval for several different products [44].

Adipose tissue is a major source of multipotent stem cells in the SVF, a heterogeneous population that includes adipose-derived stem cells (ADSCs), hematopoietic cells, endothelial progenitor cells, and pericytes [45,46,47,48,49]. This heterogeneous pool of cells can be used in surgery, taking advantage of the immunomodulatory and pro-angiogenetic activities of the different cell populations [50,51,52,53]. ADSCs can differentiate into multiple cell types and have marked paracrine activity. However, the use of ADSCs requires additional ex vivo steps that require several days and involve a risk of exposure [54]. This limitation can be overcome by using SVF, which is readily extracted from lipoaspirates following enzymatic digestion. The heterogeneous pool cell subsets recapitulate the composition of the adipose tissue microvasculature and improve the immunomodulatory and angiogenic effects synergistically. According to reports such as the European Directive No. 4694/2007, autologous SVF is considered as an advanced therapy medicinal product that is extracted using enzyme digestion and is not intended to be used for the same essential functions in the recipient and donor [55]. Here, we tried to use autologous adipose tissue directly without enzyme treatment by obtaining a small volume of tissue during mastoidectomy and implanting it directly, minimizing external contact or any kind of manipulation. We examined whether the SVF and the ADSCs present in these adipose tissues could differentiate into bone and achieve bone regeneration.

Platelet-rich plasma is an autologous biologic containing platelets currently used in orthopedics for regenerating bone, cartilage, tendon, and ligaments. PRP is obtained from peripheral venous blood and contains many growth factors, including vascular endothelial growth factor (VEGF), platelet-derived growth factor (PDGF), and fibroblast growth factor-2 (FGF-2) [56,57]. These factors stimulate osteoblastic progenitor cell proliferation and differentiation. FGF-2 and VEGF also enhance tissue revascularization and angiogenesis [58]. Numerous studies have suggested that the use of PRP in the osteogenic environment in wounds increases osteoblast proliferation, extracellular matrix production, and fibroblastic differentiation. However, PRP preparation protocols are not standardized in terms of the growth factor release activation method, anticoagulant use, centrifugation conditions, or the volume of whole blood used, and the protocols have significantly different outcomes. For instance, anticoagulants commonly used include ethylenediaminetetraacetic acid (EDTA), sodium citrate, and acid-citrate-dextrose. The centrifugation steps can include an initial soft spin and a second hard spin with variable times and speeds. Platelets can be activated via freeze–thaw cycles to disrupt the platelet membrane and release growth factors, or exogenously using either calcium chloride or thrombin. Several researchers recommend avoiding the use of thrombin because it can result in myocardial infarction and peripheral blood clots. Moreover, bovine thrombin can cross react with human factor V leading to coagulopathies [59].

Another factor that contributes to PRP variability is the concentration of other cell types, such as leukocytes. It is debated whether leukocytes should be retained in PRP and the results of studies comparing leukocyte-rich versus pure (or leukocyte-poor) PRP are confounding. Filardo et al. reported that leukocyte-rich PRP resulted in a higher incidence of adverse effects when treating osteoarthritis than pure PRP, which had a lower leukocyte concentration [60]. Because the leukocytes in PRP are sources of pro-inflammatory mediators, pro-inflammatory cytokines such as interleukin-1β and tumor necrosis factor-α are activated and secreted, resulting in reduced synthesis of both COL1A1 and COL3A1 and the production of destructive proteases that inhibit extracellular matrix formation and promote its degradation. This suggests that minimizing leukocytes in PRP is more important than maximizing the platelet concentration in terms of decreasing catabolic metabolic factors and inducing matrix formation gene expression. Similarly, Oryan et al. reported that high concentrations of leukocytes in PRP appear to interfere with bone regeneration by inducing an inflammatory response that may become persistent, whereas low concentrations of leukocytes may not be able to induce the requisite inflammatory response needed in early bone regeneration [61]. Given these conflicting reports, the effect of leukocyte content on bone healing is still unclear and merits investigation. Consequently, the current study evaluated the osseous regeneration efficacy of WP obtained by a single centrifugation step for separating unneeded erythrocytes, while simultaneously retaining suitable numbers of platelets that are not excessive.

Clinically, using autologous adipose tissue and autologous WP to promote osteogenic differentiation intraoperatively are advanced therapies that do not need additional manipulations before implantation. When combined with a PCL scaffold, they had sufficient mechanical and osteoinductive properties, as in prior studies [62,63,64,65,66,67,68,69].

## 5. Conclusions

In summary, the results of this study demonstrated that the combination of rSVFs with a PCL scaffold and autologous WP can remarkably accelerate osteogenesis in a rat mastoid bone defect model. We evaluated bone formation by micro-CT imaging, Alizarin Red S staining, and immunohistochemistry. Comparing PPP, PRP, and WP, in vivo and in vitro, the whole plasma exhibited a greater ability to promote osteogenic induction. This approach represents a promising and efficient clinical technique for the anatomical and functional reconstruction of postoperative temporal bone defects following mastoidectomy.

## Figures and Tables

**Figure 1 polymers-14-00877-f001:**
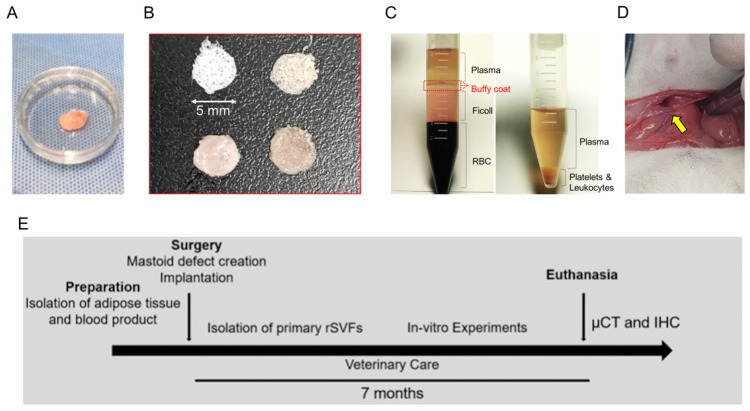
In vivo study design. The experimental composites included (**A**) uncultured stromal cells (fat) in a 35 mm dish, (**B**) PCL scaffold with blood products, and (**C**) blood products for bone reconstruction in mastoid bulla (15 mL conical tube). (**D**) Surgical procedure was performed with an anterior auricular incision with the yellow arrow indicating mastoid bulla defects. (**E**) The overall study schedule of the experiment for mastoid regeneration.

**Figure 2 polymers-14-00877-f002:**
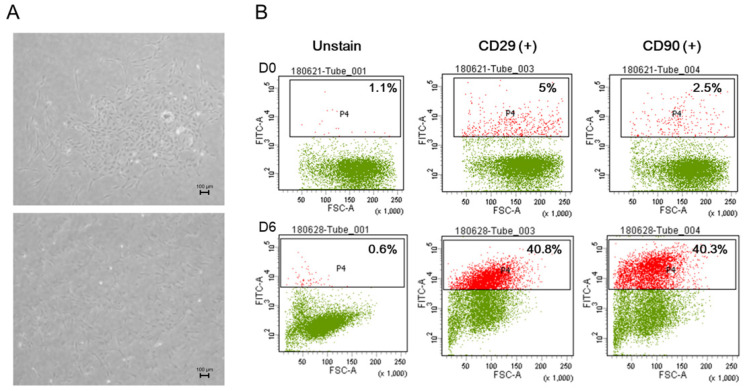
In vitro characterization of uncultured stromal vascular fraction cells isolated from rat adipose tissue. (**A**) Phase-contrast image on day 5, passage 0 and day 1, passage 1. (**B**) Cell surface markers CD29 and CD 90 showed the stemness of the rat stromal vascular fraction cells from day 0, with outstanding results by day 6. This indicates the high potential of rSVFs as an efficient source for bone differentiation. Scale bar: 100 µm, magnification: ×100.

**Figure 3 polymers-14-00877-f003:**
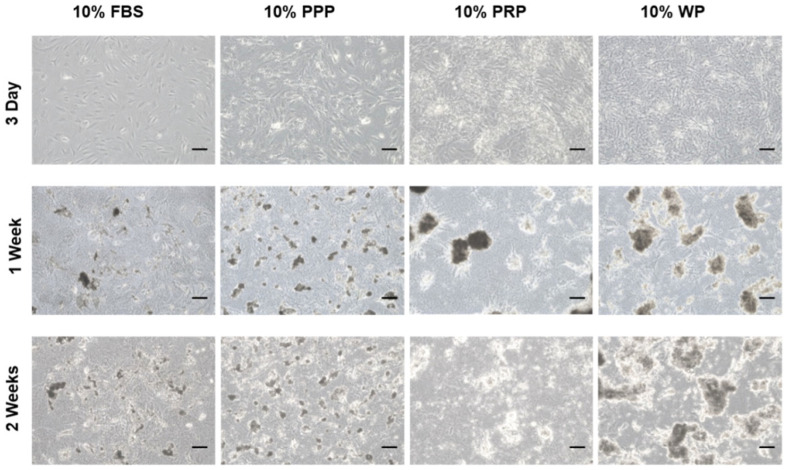
ECM mineralization of the cultured rSVFs at passage 1. Optical microscopic images on day 3, 1 week, and 2 weeks showed the most mineralization in the rat stromal vascular fraction cells treated with 10% (*v*/*v*) WP. PPP: platelet-poor plasma, PRP: platelet-rich plasma, WP: whole plasma, scale bar: 100 µm, magnification: ×100.

**Figure 4 polymers-14-00877-f004:**
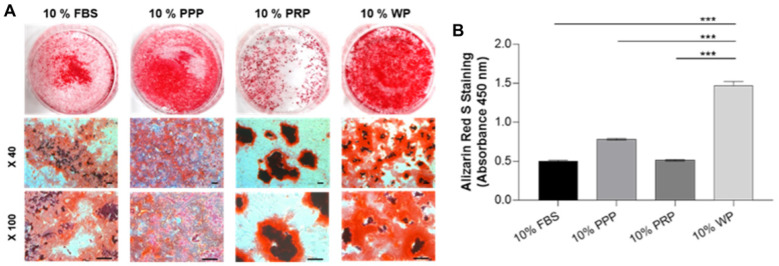
Semi-quantitative analysis of Alizarin Red S staining in cultured rSVFs at passage 1. (**A**) Optical microscopic images at 3 weeks showed the most mineralization in the SVF treated with WP and (**B**) quantitative results of retention of Alizarin Red S corresponding to the previous phenotype data, PPP: platelet-poor plasma, PRP: platelet-rich plasma, WP: whole plasma, scale bar: 200 µm, *** *p* < 0.001 by Mann–Whitney *U* test.

**Figure 5 polymers-14-00877-f005:**
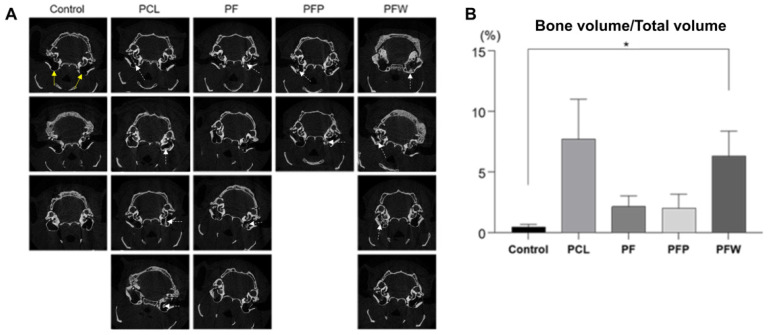
Measurement of newly formed bone by micro-CT. (**A**) After 7 months of follow up, bone tissue densities in SD rat mastoid cavities were observed by micro-CT. White dotted arrows are where the bone densities are relatively well observed. (**B**) The quantitative analysis of bone volume/total volume ratio represented that the PCL scaffold + fat + WP group had the highest bone volume/total volume ratio with a significant difference compared to the control group (defect). The yellow arrows indicate mastoid bulla defects. (Control group: defect only (*n* = 3); PCL group: PCL 300 mg c 20 wt% without fat (*n* = 4); PCL + fat group: PCL 300 mg c 20 wt% and fat 50 mg (*n* = 4); PCL + fat + PRP group: PCL 300 mg 20 wt%, fat 50 mg, and PRP 50 uL (*n* = 2); PCL + fat + WP group: PCL 300 mg c 20 wt%, fat 50 mg, and WP 50 uL (*n* = 4)), PRP: platelet-rich plasma, WP: whole plasma, * *p* < 0.05 by Mann–Whitney *U* test.

**Figure 6 polymers-14-00877-f006:**
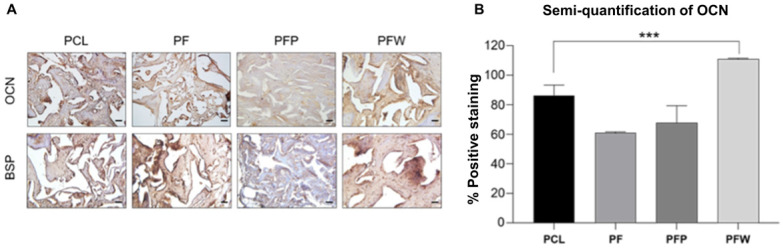
New bone formation of engineered PCL scaffolds at 7 months of implantation. (**A**) Immunohistochemical staining of osteocalcin and bone sialoprotein (DAB-hematoxylin, bright field) and (**B**) semi-quantification of OCN expression measured by ImageJ software. PCL group: PCL 300 mg c 20 wt% neither fat nor blood products (*n* = 4); PF (PCL + fat) group: PCL 300 mg c 20 wt% and fat 50 mg (*n* = 4); PFP (PCL + fat + PRP) group: PCL 300 mg 20 wt%, fat 50 mg, and PRP 50 μL (*n* = 2); PFW (PCL + fat + WP) group: PCL 300 mg c 20 wt%, fat 50 mg, and WP 50 μL (*n* = 4)), scale bar: 50 µm, magnification: ×200, *** *p* < 0.005 by ANOVA and post hoc tests.

## Data Availability

Not applicable.

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
