# Peer review of "Development of Intracorporeal Differentiation of Stem Cells to Induce One-Step Mastoid Bone Reconstruction during Otitis Media Surgeries"

_polymers, 2022, doi:10.3390/polym14050877_

Round 1

Reviewer 1 Report

Major:

  1. What does those numbers (e.g. 44074, 401757 etc.) mean in the section of 2.1?
  2. Line 103, “the scaffolds were immersed in distilled water to remove NaCl……”. How long does this immersion process take?
  3. Figure 1A-D, please add scale bars;
  4. Figure 5A and Figure 5B were not indicated in the corresponding text;
  5. It was indicated that “…… groups had relatively small amounts of bone formed……” (line 267), could the authors help the readers to identify the bone formed from the Micro-CT images?
  6. For Figure 5A, why some groups have 4 representative images, while others have 3 or 2 representative images?

Minor:

  1. Line 97, it shall be “Fabrication of the PCL scaffold”;
  2. Please check and provide the company names and addresses (city, country) of the instruments, agents and software used. For USA or Canadian companies, please all provide the state name, i.e., city name, abbreviated state name, USA/Canada).

Reviewer 2 Report

The manuscript titled Development of intracorporeal differentiation of stem cells to induce one-step mastoid bone reconstruction during otitis media surgeries is well written and scientifically sound. The authors can do the minor changes and explain my doubts.

  1. Why PRP used instead of PRF?
  2. What was the negative and positive control used for AZ staining and Osteogenic differentiation.
  3. What was the rational for using exactly 7 months as the time point for in vivo experiments?
  4. Why was OCN alone used for IH? Why not other early markers for bone formation like RunX2?
  5. The authors can add the reference for enhancing the manuscript

"Mozafari M, editor. Handbook of Biomaterials Biocompatibility. Woodhead Publishing; 2020 Jun 17".

Round 2

Reviewer 1 Report

The reviewer believe it would be necessary for the authors to explain why they use different number of rats for the control group, the PCL group, PFP group and the PFW group.
